# Acteoside Ameliorates Hepatic Steatosis and Liver Injury in MASLD Mice Through Activation of PINK1/Parkin-Related Mitophagy Markers

**DOI:** 10.3390/nu18010118

**Published:** 2025-12-29

**Authors:** Meili Cong, Xinxin Qi, Hongguang Sun, Xinxuan Zhang, Yunxin Yan, Tao Liu, Jun Zhao

**Affiliations:** 1School of Public Health, Xinjiang Medical University, Urumqi 830017, China; congmeili@126.com (M.C.); xjmuqxx@163.com (X.Q.); sunhongguang78@aliyun.com (H.S.); 18835454567@163.com (X.Z.); 19025782260@163.com (Y.Y.); 2Animal Laboratory Center, Xinjiang Medical University, Urumqi 830017, China; 3Xinjiang Key Laboratory for Uighur Medicine, Institute of Materia Medica of Xinjiang, Urumqi 830004, China

**Keywords:** liver steatosis, acteoside, lipid accumulation, PINK1/Parkin pathway

## Abstract

**Objective**: Acteoside (ACT) has different pharmacological properties such as antioxidant, hepatoprotective and anti-inflammatory effects. Impaired mitophagy has been recognized as an important pathogenic factor in metabolic dysfunction-associated steatotic liver disease (MASLD). Nevertheless, the possible therapeutic role of ACT in MASLD and the exact effect of ACT on mitophagy regulation are not explored. This study aims to elucidate the therapeutic efficacy of ACT in a high-fat and high-sugar (HFHS) diet-induced mouse model of MASLD and to determine whether its effects are related to the activation of PINK1/Parkin-related mitophagy markers. **Methods**: C57BL/6J mice were randomly allocated to control, model, rosuvastatin (RSF, 3 mg/kg), and ACT (30, 60, and 120 mg/kg) groups. Following a 14-week continuous intervention, biochemical parameters, liver histology, and mitophagy-related markers were assessed. **Results:** ACT administration significantly improved serum lipid profiles, liver function and insulin resistance, marked by reduced levels of MDA, IL-6, TNF-α, IL-1β, LDL-C, TC, TG, AST, ALT, HOMA-IR (*p* < 0.05), while increasing HDL-C and enhancing hepatic GSH-Px and SOD activities (*p* < 0.05). Histological examination revealed a notable attenuation of hepatic steatosis and lipid accumulation. At the molecular level, ACT promoted mitophagy activation, as indicated by upregulated PINK1, LC3II/I, and Parkin expression and downregulated P62 and p-P62. Electron microscopy further validated the restoration of mitochondrial morphology and reduction in lipid droplets. **Conclusions:** These results demonstrate that ACT ameliorates MASLD progression by improving metabolic homeostasis, reducing inflammation and oxidative stress, and alleviating PINK1/Parkin-related mitophagy impairment to restore mitophagy homeostasis. Our study highlights the potential of ACT as a new therapeutic agent for MASLD.

## 1. Introduction

Metabolic dysfunction-associated steatotic liver disease (MASLD), with a global prevalence of approximately 33%, represents the most common chronic liver disease. If not treated promptly, it can progress from fibrosis to cirrhosis and ultimately to hepatocellular carcinoma, posing a major threat to global health [1,2]. Excessive intrahepatic lipid accumulation is a primary pathological manifestations of MASLD [2,3]. Mitochondria serve as central organelles in lipid metabolism, and their dysfunction has been established as a critical factor in MASLD pathogenesis. Mitophagy, a selective autophagic process responsible for the clearance of impaired mitochondria, is essential for sustaining mitochondrial homeostasis; its impairment represents an early hallmark of MASLD [4]. It was reported that PINK1/Parkin pathway functions as a central mediator of mitophagy and exerts essential control over this quality control mechanism [5]. In MASLD, dysfunction of the PINK1/Parkin-driven mitochondrial clearance pathway leads to the accumulation of impaired mitochondria, excessive ROS generation, disruption of mitochondrial dynamics, dysregulated lipid metabolism, oxidative stress (OS) as well as activation of inflammatory responses. These changes exacerbate hepatocyte injury and promote disease progression [6]. Therefore, targeting PINK1/Parkin pathway and restoring mitophagy balance represents a promising therapeutic strategy for MASLD [7,8].

Acteoside (ACT), a naturally occurring phenylethanoid glycoside, is present in a wide variety of medicinal plants, such as *Cistanche tubulosa*, *Rehmannia glutinosa*, and *Verbena officinalis* [9,10,11]. ACT possessed multiple pharmacological properties, including anti-inflammatory, anti-fatigue, antioxidant, neuroprotective, and hepatoprotective activities [12,13,14,15]. ACT was found to effectively alleviate hepatic lipid accumulation in diabetic mice via activation of the Nrf2/HO-1 pathways [16]. Previous studies have shown that ACT diminishes the hepatic stellate cell activation and alleviates liver fibrosis, thereby conferring protective effects on hepatocytes. However, whether ACT improves MASLD through activation of the mitophagy markers and the precise molecular mechanisms remain to be clarified. Therefore, this research was designed to elucidate whether the protective effect of ACT against MASLD in mice is associated with the regulation of PINK1/Parkin pathway to maintain mitochondrial homeostasis. The results may offer experimental support for the advancement of ACT-based hepatoprotective agents as well as treatment and prevention of MASLD.

## 2. Materials and Methods

### 2.1. Chemicals and Reagents

Acteoside (ACT, ≥98% purity) was sourced from Chengdu Manst Biotechnology Co., Ltd. (Chengdu, China). Rosuvastatin was sourced from AstraZeneca Pharmaceuticals (China) Co., Ltd. (London, UK). High-fat diet and fructose were sourced by Jiangsu Xietong Biotechnology Co., Ltd. (Nanjing, China) and Shanghai Yuanye Biotechnology Co., Ltd. (Shanghai, China), respectively. Commercial assay kits for determining the levels of ALT, AST, TG, TC, HDL-C, LDL-C, TNF-α, IL-1β, IL-6, SOD, MDA, and GSH-Px were acquired from Shenzhen Mindray Bio-Medical Technology Co., Ltd. (Shenzhen, China) and Wuhan Elabscience Biotechnology Co., Ltd. (Wuhan, China). Additional reagents, including electron microscope fixative was sourced from Wuhan Saiwell Biotechnology Co., Ltd. (Wuhan, China), along with the BCA protein quantification kit, RIPA lysis buffer, SDS-PAGE gel preparation kit, and ECL developer, were acquired from Beijing Pulley Gene Technology Co., Ltd. (Beijing, China). Primary antibodies targeting PINK1, Parkin, LC3, P62, and p-P62 were procured from Affinity Biosciences Group Ltd. (Changzhou, China).

### 2.2. Animal Experiments

Specific pathogen-free (SPF) C57BL/6J male mice (18–22 g) were obtained from the Animal Laboratory Center of Xinjiang Medical University (Production License No. SCXK [Xin] 2023-0001). We chose the strain and sex in line with our previous study and common practice in metabolic liver disease research. This model exhibits a robust development of diet-induced obesity and fatty liver phenotypes, exhibiting shorter modeling duration, higher modeling consistency, and lower variability compared to females [17,18]. The animals were kept in a controlled barrier environment at the Animal Laboratory Center of Xinjiang Medical University (Use License No. SYXK [Xin] 2023-0004). The laboratory was sustained at a constant temperature of 22 to 25 °C, with a relative humidity of 50 to 60% and a 12 h light/dark cycle. During the experimental period, the mice had ad libitum access to diet and water, and the bedding was changed every three days. All animal experiments were carried out in strict agreement with the ethical guidelines issued by the Ethics Committee of Xinjiang Medical University (IACUC-JT-20240415-58).

After a week-long acclimatization phase, 48 C57BL/6J mice were divided randomly to 6 experimental groups (n = 8): control, model, rosuvastatin (RSF, 3 mg/kg) and ACT (30, 60, and 120 mg/kg) groups. We selected the group size of n = 8 was based on our preliminary experimental data and common practice in similar animal studies of metabolic liver disease, while also adhering to the principles of the 3Rs (Replacement, Reduction, Refinement) in animal research. This sample size is consistent with previously published studies utilizing comparable experimental designs [17,19]. Throughout the study, the control group was maintained on a standard chow diet, while the other groups were kept on a HFHS diet containing 16.5% protein, 25.5% carbohydrates, 58% fat, and 20% fructose in drinking water throughout the experimental period. Mice in RSF or ACT group were administered oral gavage at the indicated dose while the control group received equal volume of normal saline. Following a 14-week intervention, all mice were euthanized via pentobarbital sodium anesthesia after fasting for 12 h. Blood samples were harvested via orbital venous sinus puncture, clotted for 2–3 h at room temperature, and then centrifuged at 3000 rpm for 15 min at 4 °C to isolate serum. The resulting serum was carefully separated and stored for subsequent biochemical analysis. Immediately after blood collection, liver tissues were dissected and weighed for the calculation of liver index. All tissues were quickly frozen in liquid nitrogen and maintained at −80 °C for following molecular and histological examination.

### 2.3. Serum Biochemical Assays

Mindray BS-240VET automated biochemical analyzer (Shenzhen, China) was used to detect the concentrations of LDL-C, HDL-C, TC, TG, AST, ALT in mouse serum.

### 2.4. Homeostasis Model Assessment of Insulin Resistance (HOMA-IR) Assays

Blood glucose was determined immediately from fresh blood using a handheld glucometer. Serum insulin levels were quantified with a commercial mouse-specific ELISA kit. All procedures conducted in strict adherence to the manufacturer’s protocol. The HOMA-IR index was then calculated from these values as follows: [fasting glucose (mmol/L) × fasting insulin (μIU/mL)]/22.5.

### 2.5. Histopathological Detection

Tissue sections from the largest left liver lobe were fixed in 4% paraformaldehyde. Subsequently, the fixed tissues were subjected to oil red O and HE staining, respectively. The histopathological evaluations, including hepatocyte injury, inflammatory infiltration, and steatosis, were performed in a blinded manner by independent specialists using a light microscope.

### 2.6. Transmission Electron Microscopy Assays

Liver tissues were immediately fixed in pre-cooled glutaraldehyde, and 1 mm^3^ tissue blocks were dissected under the fixative. These tissue blocks were then transferred into a new EP tube containing fresh fixative for additional fixation at 4 °C. Subsequently, the tissue was fixed again utilizing 1% osmium tetroxide in the dark for 2 h in the dark, then dehydrated through an ascending concentration gradient of ethanol and acetone, and embedded in Epon 812 embedding medium, and polymerized. Ultrathin sections (70 nm thick) were prepared, double-stained sequentially with 2% uranyl acetate and 2.6% lead citrate. The samples were examined under a TEM (HT7800, Hitachi, Tokyo, Japan). The number and morphological integrity of mitochondria, autophagosomes, and mitophagic structures were evaluated, and representative images were captured and analyzed.

### 2.7. OS and Inflammatory Factors Detection

The levels of OS markers (GSH-Px, SOD, MDA) were measured in serum using commercial kits per the manufacturer’s directions. Liver tissues were homogenized in PBS (1:9, *w*/*v*). After protein quantification by BCA method, the levels of IL-1β, TNF-α, and IL-6 in the homogenate were determined utilizing specific ELISA kits as per the manufacturer’s protocols.

### 2.8. Western Blot (WB) Assays

Total protein was extracted from liquid nitrogen-homogenized liver tissues utilizing a BCA assay kit. After quantification against a standard curve, proteins were denatured, separated by SDS-PAGE, which were then transferred to PVDF membranes. Moreover, the membranes were then blocked with 5% skim milk and probed with primary antibodies (PINK1, Parkin, LC3, P62, p-P62) at 4 °C for the entire night, followed by incubation with HRP-conjugated secondary antibodies. Signals were developed with ECL, exposed to X-ray film, and quantified in a blinded manner by independent specialists using ImageJ (v1.46r) with GAPDH normalization.

### 2.9. Immunohistochemistry (IHC) Assays

Liver tissues were fixed in 4% paraformaldehyde, embedded in paraffin, and sectioned at 4 μm. Sections were deparaffinized, rehydrated, and antigen-retrieved in sodium citrate buffer (pH 6.0). Moreover, endogenous peroxidase was blocked with H_2_O_2_, followed by normal goat serum to reduce nonspecific binding. Primary antibodies (PINK1, Parkin, LC3, P62, p-P62) were incubated at 4 °C overnight, and HRP-conjugated secondary antibodies were applied for 1 h. DAB was used for visualization, and hematoxylin for counterstaining. Sections were dehydrated, cleared, and mounted. The quantitative analysis of target protein expression was blindly performed by independent specialists using ImageJ software (Version 1.46r), with GAPDH as the internal reference.

### 2.10. Statistical Analysis

All statistical analyses were performed using SPSS version 26.0 (IBM Corp., Armonk, NY, USA). The experimental unit for all analyses was the individual mouse, with group sizes of n = 8 for biochemical and histopathological assessments and n = 3 for WB and IHC analysis. Although the sample size of n = 3 for the latter analyses is relatively small and partly reflects funding limitations, it was determined to be statistically adequate for this experimental context. Data are report as mean ± standard deviation. Moreover, normality was tested utilizing the Shapiro–Wilk test and homogeneity of variances was assessed with Levene’s test. For normally distributed data with equal variances, one-way ANOVA followed by Tukey’s post hoc test was used for comparisons of the multiple groups. When data did not meet normality or homogeneity assumptions, the Kruskal–Wallis test with Dunn’s post hoc test was utilized (ALT, IL-1β and MDA). Statistical significance was defined as *p* < 0.05.

## 3. Results

### 3.1. Protective Effect of ACT Against Hepatic Steatosis in MASLD Mice

As revealed in Figure 1, following the 14-week HFHS diet induction, mice in the model group exhibited substantially higher body weight and liver index relative to the control group (*p* < 0.01). Treatment with ACT (30, 60, and 120 mg/kg), as well as rosuvastatin (RSF, 3 mg/kg), reduced body weight relative to the model group, with the most pronounced reduction noted in the ACT 120 mg/kg group (*p* < 0.01, Appendix A). Compared to the control group, the model group had significant increases in serum TG, TC, and LDL-C levels (*p* < 0.01) and a substantial reduction in HDL-C levels (*p* < 0.01). Intervention with ACT (30, 60, and 120 mg/kg) reduced the increased serum TG, TC, and LDL-C levels (*p* < 0.05) and normalized HDL-C levels (Appendix A). In addition, serum ALT and AST levels were substantially higher in the model group than in the control group, while treatment with ACT (30, 60, and 120 mg/kg) significantly improved these increases (*p* < 0.01, Figure 1 and Appendix A). These findings indicated that ACT could be effective in reducing hepatic lipid accumulation and improving HFHS diet-induced MASLD mice.

### 3.2. Effects of ACT on HOMA-IR in MASLD Mice

As revealed in Figure 2, HFHS feeding markedly increased fasting blood glucose and fasting insulin levels compared with the control group (*p* < 0.01). ACT treatment significantly attenuated this increase (*p* < 0.05 vs. model group).

To further evaluate systemic insulin resistance, we calculated the HOMA-IR index. The HOMA-IR value was profoundly higher in the model group than in the control group (*p* < 0.01), indicating severe insulin resistance. Importantly, ACT administration effectively ameliorated insulin resistance, as demonstrated by a significant reduction in HOMA-IR (*p* < 0.01, Appendix A).

### 3.3. Effects of ACT on Liver Pathological Changes in MASLD Mice

Macroscopic evaluation of liver samples as revealed in Figure 3 revealed that the control group showed typical reddish-brown coloration, soft texture, and sharp edges. In comparison, the model group was a pale-yellow discoloration with distinct yellow spots, severely altered texture, and blunt and rounded edges. After the treatment of ACT (30, 60 and 120 mg/kg) and RSF, the color and texture of the liver of mice were improved to varying degrees (see Appendix A for details).

Histopathological examination with H&E staining showed significant hepatic damage in the model group, including the deposition of lipids in hepatocytes, disorganized hepatic cord architecture, severe steatosis, multiple fat vacuoles, and inflammatory cell infiltration. These findings are consistent with the characteristic histopathological features of MASLD, and suggest successful induction of this disease model. Administration of ACT (30, 60, and 120 mg/kg) markedly alleviated hepatic steatosis and lipid accumulation, as evidenced by a substantial decrease in the lipid droplets’ size and number (Appendix A). Notably, the ACT (120 mg/kg) group demonstrated the most pronounced therapeutic effect. These results indicate that ACT treatment confers hepatoprotective benefits against HFHS-induced MASLD in mice. Oil Red O staining further confirmed increased lipid accumulation and fat vacuolar degeneration in the livers of the model group, whereas the ACT (30, 60, 120mg/kg) groups significantly decreased the number and volume of intracellular lipid accumulations (Appendix A). Collectively, these data indicate that ACT attenuates hepatocellular injury and reduces hepatic lipid deposition.

### 3.4. Effects of ACT on Liver Tissue Ultrastructure in MASLD Mice

Observed under transmission electron microscopy (TEM), as revealed in Figure 4, the MASLD mice had severe mitochondrial dysfunction as follows: structural disruption, membranes unclear or even dissolved, cristae disappeared, the number of incomplete mitochondria increased significantly, large amounts of lipid droplets deposited inside, and autophagosome structures. ACT (120 mg/kg) could significantly reverse these damages (Appendix A). The results indicated that ACT may maintain mitochondrial network homeostasis and function by promoting the mitochondrial autophagy process and timely removing damaged mitochondria.

### 3.5. Effects of ACT Against OS and Inflammatory Factors in MASLD Mice

To investigate the underlying mechanism by which ACT modulates hepatic mitophagy, key OS markers were measured. As illustrated in Figure 5, relative to the control group, the model group exhibited a marked reduction in SOD and GSH-Px activities, along with a marked raise in MDA level (*p* < 0.01). Following ACT treatment, these OS parameters were significantly ameliorated (*p* < 0.05, Appendix A). Our findings illustrate that the hepatoprotective effect of ACT is mediated, in part, by its antioxidant effect, with the consequent reduction of OS being correlated with the improvement of mitophagy and the maintenance of mitochondrial homeo-stasis.

Excessive hepatic lipid accumulation is a known trigger for inflammatory responses. To evaluate the ACT’s anti-inflammatory activity, we measured the levels of key pro-inflammatory cytokines in mice with HFHS diet-induced MASLD. As indicated in Figure 5, the model group displayed a substantial upregulation, IL-1β, TNF-α, and IL-6 in liver tissue homogenates relative to the control group (*p* < 0.01). In contrast, ACT treatment (30, 60, and 120 mg/kg) lowered the concentrations of these cytokines, with the most pronounced suppression observed at the 120 mg/kg dose (*p* < 0.05, Appendix A). These results demonstrate that the amelioration of MASLD by ACT is closely associated with the inhibition of pro-inflammatory cytokine expression. In conclusion, the results indicate that alleviating OS and inhibiting inflammation are the key downstream protective effects brought about by ACT enhancing mitochondrial autophagy.

### 3.6. ACT’s Effects on the Expression of Mitophagy-Related Proteins Parkin, LC3, PINK1, P62, and p-P62 in MASLD Mice

We sought to investigate the mechanism by which ACT enhances mitochondrial autophagy and ameliorates HFHS diet-induced lipid metabolism dysregulation through modulation of expression levels of PINK1, Parkin, LC3, p62, and p-p62. As shown in Figure 6, relative to the control group, the model group had significantly lower levels of PINK1, Parkin, and LC3II (*p* < 0.01) and higher levels of p62 and p-p62 (*p* < 0.01) in the liver. Compared to the model group, ACT treatment significantly increased PINK1, Parkin, and LC3II expression and decreased p62 and p-p62 expression (*p* < 0.05, Appendix A). This indicates that ACT attenuates hepatic lipid deposition by activating the PINK1/Parkin signaling pathway to promote mitophagy.

To further verify the above results, we performed immunohistochemistry on liver tissue samples. As shown in Figure 7, extensive positive signals for PINK1, Parkin, and LC3 (brown-yellow staining) could be observed in the control group, indicating that liver cells have an active mitochondrial quality control mechanism under physiological conditions. In contrast, the model group showed a marked reduction in both staining intensity and positive cell rate for PINK1, Parkin, and LC3 compared with the control group (*p* < 0.01). Conversely, the staining intensity and positive cell rate for the autophagic substrates P62 and p-P62 were substantially improved in the model group (*p* < 0.01). ACT administration significantly increased the expression of PINK1, Parkin and LC3 proteins (*p* < 0.05, Appendix A). These results suggest that ACT improves hepatic PINK1, Parkin, and LC3 expression and represses P62 and p-P62 expression in MASLD mice. Together with its ameliorative effects on markers of OS, these results indicate that ACT may improve mitochondrial function and reduce oxidative damage by activating PINK1/Parkin-related mitophagy markers, which may be a key mechanism by which it preserves mitochondrial homeostasis and ameliorates MASLD.

## 4. Discussion

MASLD is a multisystem disorder and is strongly linked with metabolic dysregulation and often coincides with other conditions such as diabetes, obesity, and hypertension. Epidemiological studies suggest that the global prevalence of MASLD is one-third, with a steady rise in its prevalence over the past few decades [20,21]. The hall mark pathological feature of MASLD is excessive accumulation of lipids in the liver and this trend has followed the widespread introduction of HFHS diets in Western populations [17,22]. Although the exact pathogenesis of MASLD is still not fully explained, a growing body of evidence highlights mitochondrial dysfunction as a key mechanism in the pathogenesis of MASLD [4,23]. Therefore, maintenance of mitochondrial homeostasis and function is thought to be critical for successful therapeutic intervention. Mitophagy, as a selective autophagy process that leads to degradation of damaged mitochondria, is a key guardian of mitochondrial quality control. Growing evidence suggests that mitophagy is implicated in the regulation of both lipolysis and lipogenesis and impairment of mitophagy contributes substantially to the MASLD’s onset and progression [24,25,26]. Currently, there are no approved pharmacotherapies for MASLD. Consequently, clinical management is based mainly on lifestyle-based strategies, with emphasis on diet improvement and regular exercise. Nevertheless, long-term patient adherence to these interventions is less than ideal [27]. Thus, developing new agents that promote mitophagy in the liver is a promising therapeutic approach.

Recently, natural active compounds have attracted more and more attention in the field of MASLD research because of their multifaceted effects such as the regulation of lipid metabolism, the suppression of inflammation, and the reduction of OS. ACT is a bioactive natural product that exhibits a wide range of pharmacological activities with a protective potential against various liver diseases [28]. Existing studies have reported that ACT has important hepatoprotective effects in models of immune-mediated liver injury and effectively inhibits hepatic stellate cell activation [12,29]. Based on these findings, we hypothesize that ACT may have potential preventive and therapeutic benefits against MASLD. The HFHS diet causes hepatic lipid deposition, intracellular abnormal accumulation of TG and TC, and systemic metabolic disturbances leading to hepatic injury, a process that closely resembles the pathogenesis of MASLD in humans and successfully recapitulates the clinical features of the disease [30]. In this research, serum levels of hepatic indicators and insulin resistance were markedly elevated in the model group. Following ACT administration, these markers of metabolic and liver damage were significantly reduced. Histopathological examination further demonstrated that ACT reduced hepatic steatosis, significantly improved liver architecture, reduced the extent and number of steatotic lesions, and improved both functional impairment and lipid accumulation. Our data represent strong evidence for the hepatoprotective properties of ACT.

Impaired mitophagy causes dysfunctional mitochondria to accumulate, causing excess ROS production. This leads to lipid peroxidation, membrane integrity and permeability and eventually hepatocyte death [31,32]. OS also stimulates hepatocytes and intrahepatic immune cells to secrete pro-inflammatory cytokines (i.e., IL-1β, TNF-α, and IL-6), which trigger hepatic inflammation. In turn, inflammation worsens OS, producing a self-perpetuating cycle that speeds up liver damage [33]. In this research, the model group had high levels of inflammatory cytokines and OS markers compared to the control group, validating the successful induction of diet-induced liver injury. ACT administration resulted in a substantial downregulation of IL-1β, TNF-α, and IL-6 expression; simultaneously, OS was significantly improved, as demonstrated by a significant decrease in MDA levels and an increase in the activities of antioxidant enzymes, such as the GSH-Px and SOD. Our findings suggest that ACT interrupts this pathological cascade by suppressing inflammation and mitigating OS, thus slowing MASLD progression.

Mitochondrial dysfunction is a principal factor in the progression of MASLD, and the accumulation of defective mitochondria from dysfunctional mitophagy is a critical link in promoting lipotoxicity, OS, and inflammation. By selectively eliminating unhealthy mitochondria, mitophagy promotes the turnover of mitochondria and maintains cellular energy homeostasis [34]. Transmission electron microscopy (TEM) showed that livers of HFHS-fed mice had pronounced mitochondrial swelling, cristae disruption and a close to complete absence of mitophagosomes, a hallmark of severe mitophagy impairment. Notably, intervention with ACT (120 mg/kg) significantly reversed these ultrastructural abnormalities.

Emerging evidence emphasizes the protective role of mitophagy in preventing tissue injury and hepatic homeostasis by the clearance of damaged mitochondria [30]. To understand the molecular mechanisms involved, we focused on the PINK1/Parkin signaling pathway, a well-characterized mitophagy regulator. WB and IHC analyses consistently showed that ACT treatment (30, 60 and 120 mg/kg) substantially upregulated PINK1 and Parkin protein expression and reduced accumulation of LC3, P62 and p-P62, indicating restored autophagic flux. Collectively, these findings suggest that ACT ameliorates MASLD, and this effect is associated with activation of PINK1/Parkin-related mitophagy markers, which improves mitochondrial quality control and ultimately reduces hepatic lipid deposition. In conclusion, ACT is a viable multi-target therapeutic candidate for MASLD.

A major strength of this study is the integrated, multi-level approach, which included the combination of in vivo assessment of phenotypic improvements with mechanistic analyses at the ultra-structural and molecular levels. Beyond recording amelioration in serum biochemical markers and histopathology, we also recorded better mitochondrial morphology by transmission electron microscopy, and confirmed changes in expression of key proteins by Western blot and immunohistochemistry. Furthermore, the pro-protective effects observed lend more support to the causal role of ACT in eliciting these benefits.

Certainly, there are several limitations of this study. Although our results provide a strong link between ACT and the activation of PINK1/Parkin-related mitophagy, a definitive causal relationship remains to be established. Further studies with genetic knockout models such as PINK1 or Parkin deficient mice would help unequivocally validate the necessity of this pathway in mediating the effects of ACT. In addition, the specific molecular target of ACT upstream of PINK1 has not yet been identified. The morphological assessment of mitochondria in this study was based on qualitative evaluation of TEM images. Although these observations support the main findings, the application of quantitative methodologies in future studies is warranted to generate more objective datasets. Finally, the translational relevance of these results is limited by inherent differences between animal models and human disease. Although the HFHS diet-induced model recapitulates some of the key features of human MASLD, it is unable to fully recapitulate the complexity and chronicity of the condition in patients. Moreover, the pharmacokinetic behavior, optimal dosing, and long-term safety profile of ACT in humans remain to be thoroughly characterized. These limitations highlight important avenues for future investigation.

## 5. Conclusions

Our study demonstrates that ACT effectively ameliorates HFHS diet-induced MASLD in mice by activating the PINK1/Parkin mitophagy markers, thus improving mitophagy and restoring mitophagy homeostasis. Our findings support the potential application of ACT in validating the PINK1/Parkin pathway as an ideal therapeutic target for novel MASLD drug development. However, the underlying mechanism is more complex than that described in this article, and our findings do not exclude the potential involvement of additional ACT-induced mechanisms in the therapeutic effects on MASLD.

## Figures and Tables

**Figure 1 nutrients-18-00118-f001:**
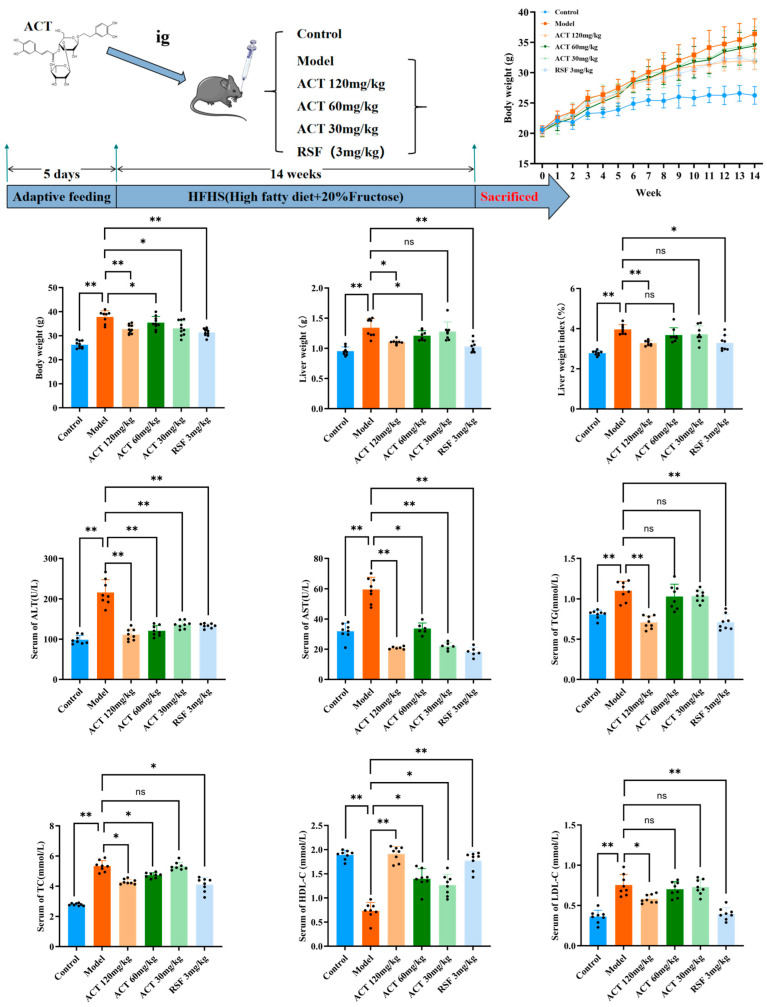
Protective effect of ACT on hepatic steatosis by improving liver function in MASLD mice. Data are expressed as means ± SD with a sample size of n = 8. Statistical significance was defined as * *p* < 0.05 and ** *p* < 0.01.

**Figure 2 nutrients-18-00118-f002:**
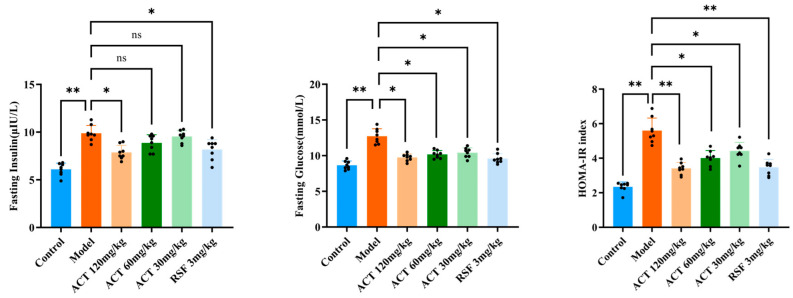
Effects of ACT on HOMA-IR in MASLD mice. Data are expressed as means ± SD with a sample size of n = 8. Statistical significance was defined as * *p* < 0.05 and ** *p* < 0.01.

**Figure 3 nutrients-18-00118-f003:**
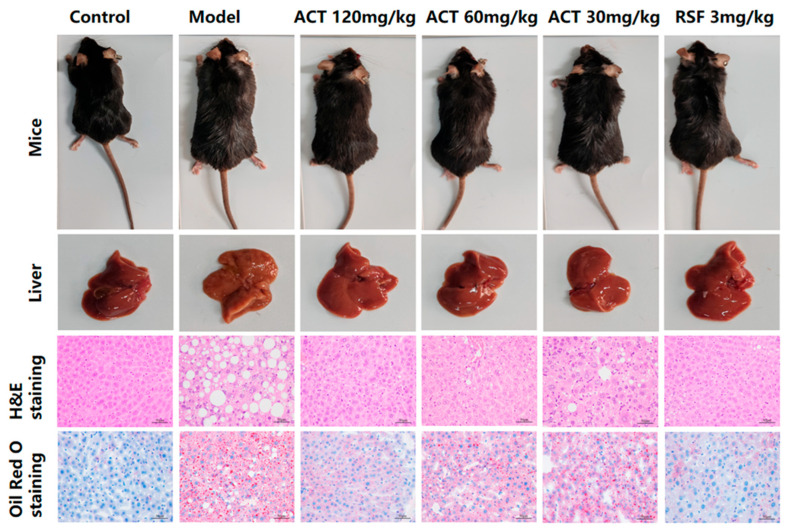
ACT alleviates liver histopathology and lipid deposition in MASLD mice (H&E and Oil Red O staining, ×400-fold).

**Figure 4 nutrients-18-00118-f004:**
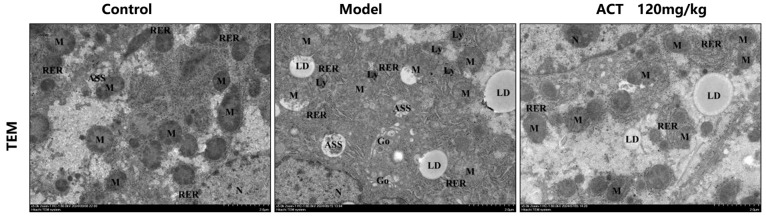
Ultrastructure of liver mitochondria observed under transmission electron microscopy (scale bar = 2 μm, 500 nm; RER: rough endoplasmic reticulum, M: mitochorome, CM: cell membrane, N: nucleolus, Go: golgi apparatus, Ly: lysosome, ASS: autolysosome, AP: autophagosome, LD: lipid droplet).

**Figure 5 nutrients-18-00118-f005:**
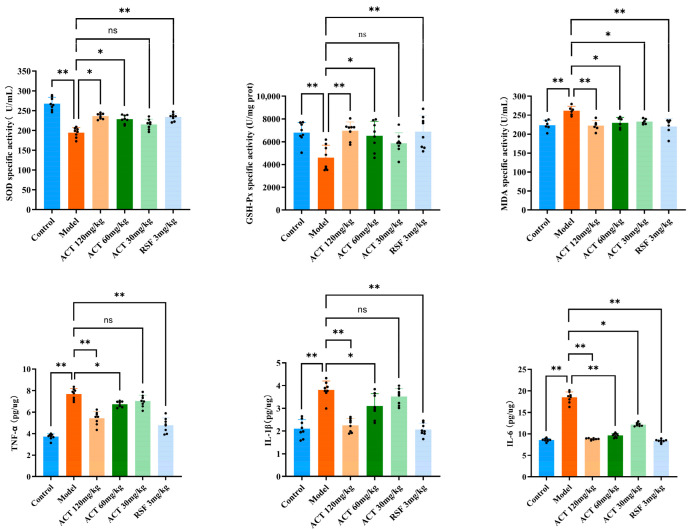
ACT’s effects against OS and inflammatory factors in the MASLD mice. Data are expressed as means ± SD with a sample size of n = 8. Statistical significance was defined as * *p* < 0.05 and ** *p* < 0.01.

**Figure 6 nutrients-18-00118-f006:**
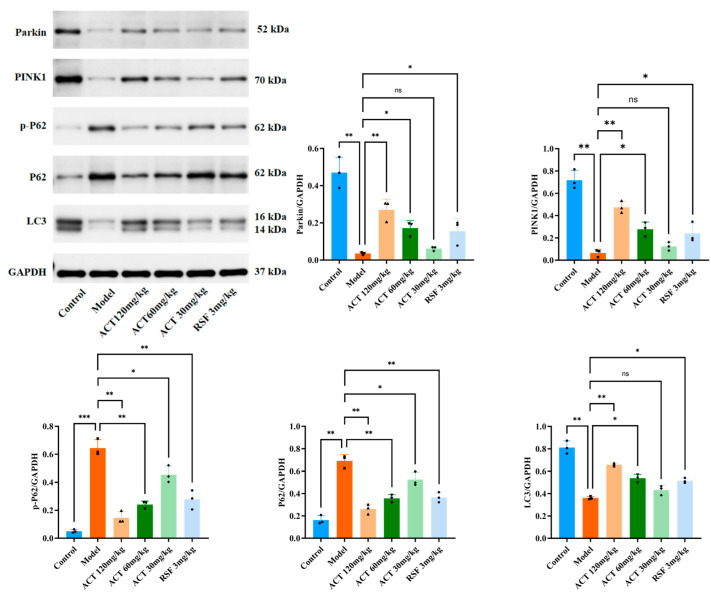
Effects of ACT on the expression of mitophagy-related proteins in liver tissue. Data are expressed as means ± SD with a sample size of n = 3. Statistical significance was defined as * *p* < 0.05, ** *p* < 0.01, and *** *p* < 0.001.

**Figure 7 nutrients-18-00118-f007:**
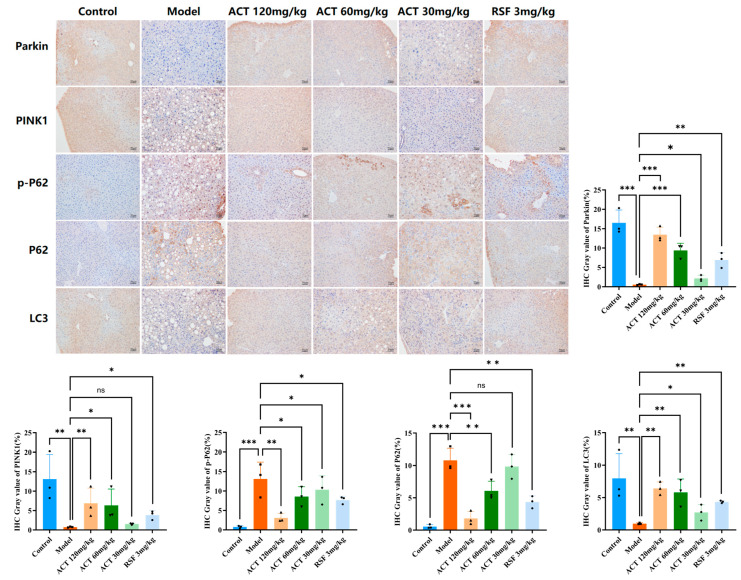
Effects of ACT on the positive immunoreactivity of mitophagy-related proteins in MASLD mice. Data are expressed as means ± SD with a sample size of n = 3. Statistical significance was defined as * *p* < 0.05, ** *p* < 0.01, and *** *p* < 0.001.

## Data Availability

The raw data supporting the conclusions of this article will be made available by the authors without undue reservation.

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
