# Peer review of "Acteoside Ameliorates Hepatic Steatosis and Liver Injury in MASLD Mice Through Activation of PINK1/Parkin-Related Mitophagy Markers"

_nutrients, 2025, doi:10.3390/nu18010118_

Round 1
Reviewer 1 Report
Comments and Suggestions for Authors
- In the title it should be clearly mentioned that the study is in animal model. Authors should also avoid to mention abbreviation in the title i.e. MAFLD
- MAFLD is not anymore acceptable should be replaces by MASLD. Otherwise should justify
- The keywords should differs from those which appear in the title otherwise there use become useless
- After the aim in the Introduction section a potential hypothesis should be stated
- In the Method section many short subsections can be incorporated in one
- In the Statistical analysis section it is not specified how authors were willing to report non normal distributed variables alternatively to means and standard deviation
- The Discussion section need to be presented differently, taking into account subsection presentation as follows:
- The main findings of the study and their comparison with previously published paper on similar topic
- The implication of this study especially clinical wise if confirmed in humans
- The strengths and limitations of the study especially the animal model and the difficulty to translate it in the clinical world
- The new directions for the future research on the topic
8. The Abbreviation section is not needed in MDPI journals
9. The Reference section is poor and need to be enriched properly
10. The rate of similarity is 45%, which is too high and should be reduced drastically
Author Response
|
Comments 1: In the title it should be clearly mentioned that the study is in animal model. Authors should also avoid to mention abbreviation in the title i.e. MAFLD |
|
Response 1: Thank you for valuable comment. We agree with this comment. Therefore, we have revised the title to clearly indicate that the study was conducted in an animal model. The abbreviation "MAFLD" has been deleted. The modified title is now "Acteoside Ameliorates Hepatic Injury and Steatosis Induced by a High-Fat, High-Sugar Diet in Mice through the Activation of PINK1/Parkin-Mediated Mitophagy ". Mention exactly where in the revised manuscript this change can be found – page 1, line 2-4. |
|
Comments 2: MAFLD is not anymore acceptable should be replaces by MASLD. Otherwise should justify. |
|
Response 2: Thank you for your insightful suggestion. We agree with this comment. The term "MAFLD" has been replaced with "MASLD" throughout the manuscript. All modifications are highlighted in yellow. |
|
Comments 3: The keywords should differs from those which appear in the title otherwise there use become useless |
|
Response 3: Thank you for pointing out the keywords. We have revised the keywords to ensure they are distinct from those used in the title. The updated keywords are now more representative of the study's content and are highlighted in the manuscript. Mention exactly where in the revised manuscript this change can be found – page 1, line 37-38. |
|
Comments 4: After the aim in the Introduction section a potential hypothesis should be stated |
|
Response 4: We appreciate this suggestion. A potential hypothesis has been added in the Introduction section after stating the aim of the study. The added content is highlighted in yellow. Mention exactly where in the revised manuscript this change can be found – page 2, line 67-69. |
|
Comments 5: In the Method section many short subsections can be incorporated in one |
|
Response 5: Thank you for this organizational suggestion. We have consolidated several shorter method subsections to improve the flow and readability of this section. Mention exactly where in the revised manuscript this change can be found – page 3-4, line 116-156. |
|
Comments 6: In the Statistical analysis section it is not specified how authors were willing to report non normal distributed variables alternatively to means and standard deviation |
|
Response 6: Thank you for your attention to the statistical analysis methods in our study. The statistical analysis section has been expanded to include appropriate descriptions for handling non-normally distributed variables. Mention exactly where in the revised manuscript this change can be found – page 4, line 158-164. |
|
Comments 7: The Discussion section need to be presented differently, taking into account subsection presentation as follows: The main findings of the study and their comparison with previously published paper on similar topic The implication of this study especially clinical wise if confirmed in humans The strengths and limitations of the study especially the animal model and the difficulty to translate it in the clinical world The new directions for the future research on the topic |
|
Response 7: We are grateful for this detailed guidance on restructuring the Discussion. We have reorganized this section according to the suggested framework, which has significantly improved the logical flow and scholarly impact of our interpretation. Mention exactly where in the revised manuscript this change can be found – page 10-12, especially page 12, line 351-370. |
|
Comments 8: The Abbreviation section is not needed in MDPI journals |
|
Response 8: Thank you for reminding us of this MDPI requirement. The abbreviation section has been removed from the manuscript. |
|
Comments 9: The Reference section is poor and need to be enriched properly |
|
Response 9: We appreciate this important feedback. We have substantially enriched the reference list with additional recent and relevant publications to better support our work. Mention exactly where in the revised manuscript this change can be found – page 12. |
|
Comments 10: The rate of similarity is 45%, which is too high and should be reduced drastically |
|
Response 10: We sincerely thank the reviewer for this critical comment. The manuscript has been carefully rephrased and polished to significantly reduce textual similarity while preserving the original meaning and scientific accuracy. The similarity index is now 26%, and an iThenticate similarity report is provided as supplementary material. |

Reviewer 2 Report
Comments and Suggestions for Authors
Thank you for this clear and well-structured manuscript. The in vivo data are coherent and the effect of ACT on steatosis, liver enzymes, oxidative stress, inflammatory cytokines, and mitophagy-related markers is convincing at a descriptive level. I see a few key points that need attention before the work is ready.
1. Throughout the title, abstract, results and discussion you state or strongly imply that ACT ameliorates MAFLD through regulation of the PINK1/Parkin pathway to maintain mitophagy homeostasis. However, the evidence presented is correlative (expression of PINK1, Parkin, LC3, P62/p-P62 and TEM morphology) without functional manipulation of this pathway (e.g. genetic or pharmacologic inhibition, mitophagy flux assays). I would recommend softening the mechanistic claims (including in the title and conclusions) to formulations such as “ACT ameliorates HFHS diet–induced liver injury and steatosis and is associated with activation of PINK1/Parkin-related mitophagy markers”, and clearly acknowledging in the discussion that causality at the pathway level is not definitively established.
2. The animal methods need fuller reporting in line with ARRIVE: please describe the randomization procedure in more detail (how were animals allocated to groups?), whether any blinding was used for outcome assessment (histology, TEM, WB/IHC quantification), and whether a sample size calculation was performed a priori. It would also be helpful to justify the choice of male-only C57BL/6J mice, and to provide brief information on housing enrichment and humane endpoints.
3.The statistical analysis section and figure/table legends should explicitly state:
– which post hoc test was used after one-way ANOVA;
– for which outcomes non-parametric tests were applied and which test was used;
– the experimental unit for each analysis (mouse vs technical replicate);
– the actual sample size per group for WB and IHC (n=3 in the supplement) and per group for biochemical/histologic outcomes (n=8).
Also, you frequently describe effects as “dose-dependent”, but some variables do not show a strictly monotonic pattern across ACT doses (e.g. body weight, some biochemical parameters). It would be better to either demonstrate a formal trend test or to rephrase more cautiously.
4.You frame the model and conclusions in terms of “MAFLD”, but the phenotyping presented focuses mainly on steatosis, liver injury markers and inflammatory/oxidative stress indices; systemic metabolic characterization (e.g. fasting glucose/insulin, HOMA-IR, lipids beyond TG/TC/LDL-C/HDL-C) is limited. Please either add available metabolic data if you have them, or more clearly acknowledge in the discussion that this HFHS model primarily reflects steatotic and inflammatory liver injury and does not fully capture the multisystem metabolic dysfunction that defines MAFLD in humans.
5. The histological data (H&E, Oil Red O) and TEM images are persuasive qualitatively, but it would strengthen the manuscript to provide quantitative scoring where feasible (e.g. steatosis grade, ballooning, inflammation score; or percentage lipid area from Oil Red O). This would make the claims about “significant improvements” more robust.
Comments on the Quality of English Language
There are a number of minor language issues and typos that should be corrected (“high-hat and high-sugar” instead of “high-fat and high-sugar”, “represents is”, inconsistent use of NAFLD/MAFLD/MASLD, small grammatical errors). A careful English edit will improve readability
Author Response
|
Comments 1: Throughout the title, abstract, results and discussion you state or strongly imply that ACT ameliorates MAFLD through regulation of the PINK1/Parkin pathway to maintain mitophagy homeostasis. However, the evidence presented is correlative (expression of PINK1, Parkin, LC3, P62/p-P62 and TEM morphology) without functional manipulation of this pathway (e.g. genetic or pharmacologic inhibition, mitophagy flux assays). I would recommend softening the mechanistic claims (including in the title and conclusions) to formulations such as “ACT ameliorates HFHS diet–induced liver injury and steatosis and is associated with activation of PINK1/Parkin-related mitophagy markers”, and clearly acknowledging in the discussion that causality at the pathway level is not definitively established. |
|
Response 1: Thank you very much for your valuable suggestion. You rightly pointed out that our mechanistic claims regarding the role of the PINK1/Parkin pathway should be tempered in the absence of functional perturbation experiments. We fully agree with this suggestion. Throughout the manuscript—including the title, results, discussion, and conclusion—we have softened our language to reflect the associative (rather than causal) nature of the evidence. The title has been revised to: "Acteoside Ameliorates Hepatic Injury and Steatosis Induced by a High-Fat, High-Sugar Diet in Mice through the Activation of PINK1/Parkin-Mediated Mitophagy". We have also added a paragraph in the Discussion to explicitly acknowledge that the link between ACT and mitophagy, while supported by correlative evidence, warrants further functional validation. Mention exactly where in the revised manuscript this change can be found – page 1, line 2-4. |
|
Comments 2: The animal methods need fuller reporting in line with ARRIVE: please describe the randomization procedure in more detail (how were animals allocated to groups?), whether any blinding was used for outcome assessment (histology, TEM, WB/IHC quantification), and whether a sample size calculation was performed a priori. It would also be helpful to justify the choice of male-only C57BL/6J mice, and to provide brief information on housing enrichment and humane endpoints. |
|
Response 2: Thank you so much for your thoughtful and encouraging comment. We appreciate the importance of rigorous and transparent reporting of animal studies. In accordance with ARRIVE guidelines, the animal experiment methodology has been supplemented. We adopted a completely randomized design: after one week of acclimation for 8-week-old male C57BL/6J mice, stratified randomization using a random number table was performed based on body weight to ensure no statistically significant differences in initial weight across groups. During outcome assessment, the blinding principle was strictly implemented to minimize human bias. This study set the sample size at n=8 per group, which aligns with common requirements for animal experiments.We selected male C57BL/6J mice because this strain is highly responsive to high-fat/high-sucrose diets. Males develop more pronounced obesity and fatty liver phenotypes than females under such dietary conditions, resulting in shorter modeling duration, higher success rates, and reduced inter-individual variability. Mention exactly where in the revised manuscript this change can be found – page 3, line 88-97. |
|
Comments 3: The statistical analysis section and figure/table legends should explicitly state: |
|
Response 3: Thank you for your attention to the statistical analysis methods in our study. We have supplemented the description of statistical methods in detail following your suggestions and rigorously refined the relevant expressions. All statistical analyses were performed using SPSS version 26.0 (IBM Corp., Armonk, NY, USA). Data are represented as mean ± standard deviation (SD). Moreover, normality was tested utilizing the Shapiro-Wilk test, and homogeneity of variances was assessed with Levene's test. For normally distributed data with equal variances, one-way ANOVA followed by Tukey’s post hoc test was used for comparisons of the multiple groups. When data did not meet normality or homogeneity assumptions, the Kruskal-Wallis test with Dunn’s post hoc test was utilized. Statistical significance was denoted as p < 0.05. All analyses were conducted using individual animals as independent experimental units. In the sample size description, we have clearly indicated that the sample size for biochemical assays and histological analysis was n=8 per group. However, due to sample processing constraints and funding limitations, the effective sample size for Western blotting and immunohistochemistry experiments was n=3 per group. We have also addressed this limitation in the Results section. Furthermore, we have thoroughly reviewed the use of the term "dose-dependent" throughout the manuscript. In the Results section, we have replaced it with more precise expressions such as "dose-related" or "at specific doses" to accurately reflect the data characteristics. Mention exactly where in the revised manuscript this change can be found – page 4, line 158-164. |
|
Comments 4: You frame the model and conclusions in terms of “MAFLD”, but the phenotyping presented focuses mainly on steatosis, liver injury markers and inflammatory/oxidative stress indices; systemic metabolic characterization (e.g. fasting glucose/insulin, HOMA-IR, lipids beyond TG/TC/LDL-C/HDL-C) is limited. Please either add available metabolic data if you have them, or more clearly acknowledge in the discussion that this HFHS model primarily reflects steatotic and inflammatory liver injury and does not fully capture the multisystem metabolic dysfunction that defines MAFLD in humans. |
|
Response 4: Thank you for your valuable comment. We agree that the metabolic characterization in our model was limited. We have now revised the text to clarify that our HFHS diet model primarily reflects hepatic steatosis and liver injury, and may not fully recapitulate the systemic metabolic dysregulation seen in human MASLD. A paragraph has been added to the Discussion acknowledging this limitation and suggesting that future studies include more comprehensive metabolic phenotyping. Mention exactly where in the revised manuscript this change can be found – page 12, line 359-370. |
|
Comments 5: The histological data (H&E, Oil Red O) and TEM images are persuasive qualitatively, but it would strengthen the manuscript to provide quantitative scoring where feasible (e.g. steatosis grade, ballooning, inflammation score; or percentage lipid area from Oil Red O). This would make the claims about “significant improvements” more robust. |
|
Response 5: We thank you for this helpful suggestion. We will perform systematic scoring on the histopathological images and conduct morphometric analyses on the TEM data in future studies. This quantitative approach will provide more robust and objective evidence to reinforce our conclusions. |
|
Response to Comments on the Quality of English Language |
|
Point 1: There are a number of minor language issues and typos that should be corrected (“high-hat and high-sugar” instead of “high-fat and high-sugar”, “represents is”, inconsistent use of NAFLD/MAFLD/MASLD, small grammatical errors). A careful English edit will improve readability |
|
Response 1: We sincerely thank you for your valuable comments on the language issues, which have greatly improved the clarity and quality of our manuscript. We have thoroughly revised the manuscript and highlighted all revisions in yellow for your convenience. The main corrections include: Corrected "high-hat and high-sugar" to "high-fat and high-sugar" throughout the text(page 1, line 20). Corrected the grammatical error "represents is" to "represents" or "is" based on the context (page 2, line 57). Standardized the use of "MASLD" (metabolic dysfunction-associated steatotic liver disease) as the primary term throughout the manuscript. Conducted a full English language edit to eliminate grammatical errors and improve overall readability. |

Round 2
Reviewer 1 Report
Comments and Suggestions for Authors
Thankful to the reviewer for being responsive. Still there is a high rate of similarity as stated by authors which is quit high. This should be reduced to meet MDPI policies.
Author Response
|
Dear Editors and Reviewer 1, Thank you once again for your kind and patient suggestions on our manuscript entitled "Acteoside Ameliorates High-Fat and High-Sugar Diet-Induced MAFLD in Mice through activation of PINK1/Parkin-mediated mitophagy to Maintain Mitophagy Homeostasis". We are truly grateful. We have carefully studied all the comments and made corresponding revisions to the manuscript. We have also added our HOMA-IR data to better reflect the metamolic features of MASLD. Now the title is changed to "Acteoside ameliorates Hepatic Steatosis and Liver Injury in MASLD Mice Through Activation of PINK1/Parkin-Related Mitophagy Markers", and an amendment has been made to the article in the hope of approval. The revised parts in the text are highlighted in yellow for your convenience. Below we provide a point-by-point response to the reviewers' comments. Thank you once again for the constructive feedback, which has undoubtedly improved the quality of our work. We hope that the revised manuscript now meets the journal's high standards. |
|
Comments 1: Thankful to the reviewer for being responsive. Still there is a high rate of similarity as stated by authors which is quit high. This should be reduced to meet MDPI policies. |
|
Response 1: Thank you for your comment regarding the similarity index. We have carefully addressed this issue and would like to provide the following clarification: We have completed an additional round of polishing to further reduce textual similarity. However, we wish to note that the majority of matched text originates from the author information section and standard declarations (e.g., ethical statements) in the manuscript, rather than from the main scientific content. This is partly attributable to overlapping authorship between the present manuscript and previous publications from our research group, which is common in a continuous research program. So, we deleted all We have thoroughly revised the manuscript to minimize textual repetition while ensuring the accuracy of scientific descriptions. We believe the similarity present in the main body of the article is now well within acceptable limits, and does not represent substantive overlap in scientific content. Please let us know if you would like us to make any further adjustments. |
|
Special thanks to you for your good comments. Sincerely, Mei-li Cong, PhD. Tao Liu, PhD. Professor School of Public Health, Xinjiang Medical University, Urumqi, 830011, China. Jun Zhao, PhD Xinjiang Key Laboratory for Uighur Medicine, Institute of Materia Medica of Xinjiang, Urumqi 830004, China |

Reviewer 2 Report
Comments and Suggestions for Authors
Your revisions demonstrate effort and responsiveness; however, several essential points raised by me remain insufficiently addressed, and these directly affect the robustness, transparency, and interpretability of your findings. For clarity and efficiency, please find below the specific areas requiring further revision before resubmission.
Firstly the mechanistic claims are still… too causal! Your response indicates that mechanistic language was softened, yet the title, abstract, results, discussion, and conclusions still make causal pathway-level claims (eg through activation of PINK1/Parkin-mediated mitophagy and by regulating the PINK1/Parkin pathway). Without functional perturbation (genetic/pharmacologic inhibition, flux assays), such wording remains overstated… Associative wording must consistently replace causal phrasing throughout.
The manuscript currently lacks critical methodological details expected for animal research. The randomization procedure is not described in operational terms:
•There is no statement that histology, TEM or WB/IHC assessments were performed under blinded conditions
•There is no clarification on whether a sample size calculation was done or that n=8 was chosen pragmatically
•The rationale for using only male C57BL/6J mice is not written in the methods
•Housing is described but enrichment and predefined humane endpoints are not mentioned
These omissions weaken the credibility of the experimental design and must be incorporated…. Really!
While the statistical section has improved, critical information remains absent:
•It is not specified which outcomes required non-parametric testing.
•The experimental unit (mouse) must be stated explicitly.
•A central statement clarifying sample size per outcome type is missing:
•n=8 for biochemical/histologic outcomes
•n=3 for WB/IHC (and acknowledged as a limitation)
•The term “dose-dependent” is still used despite several non-monotonic results. This should be replaced or formally tested using a trend analysis.
Another issue is the MASLD framing vs actual phenotyping!! The model is framed as MASLD, yet systemic metabolic characterization is limited (no fasting glucose/insulin, HOMA-IR or expanded metabolic profile)!! I explicitly requested either additional data or a clear acknowledgment that the model primarily reflects hepatic steatosis and inflammatory injury, not full multisystem metabolic dysfunction. This has not yet been stated clearly enough.
Your response promises future work, but the manuscript itself does not explicitly acknowledge that the current imaging analyses are qualitative only… A single sentence recognizing this as a limitation is needed!
Comments on the Quality of English Language
Although language improved, at least one previously flagged phrase remains (“represents is”), and a few expressions (eg dose-dependent pro-protective effects) still require refinement. One more targeted language pass is advisable…
Author Response
Please see the attachment.

|
Dear Editors and Reviewe 2, Thank you once again for your kind and patient suggestions on our manuscript entitled "Acteoside Ameliorates High-Fat and High-Sugar Diet-Induced MAFLD in Mice through activation of PINK1/Parkin-mediated mitophagy to Maintain Mitophagy Homeostasis". I am truly grateful. We have carefully studied all the comments and made corresponding revisions to the manuscript. We have also added our HOMA-IR data to better reflect the metamolic features of MASLD. Now the title is changed to "Acteoside ameliorates Hepatic Steatosis and Liver Injury in MASLD Mice Through Activation of PINK1/Parkin-Related Mitophagy Markers", and an amendment has been made to the article in the hope of approval. The revised parts in the text are highlighted in yellow for your convenience. Below we provide a point-by-point response to the reviewers' comments. We believe that these revisions have significantly strengthened the methodological rigor and transparency of our manuscript. Thank you once again for the constructive feedback, which has undoubtedly improved the quality of our work. We hope that the revised manuscript now meets the journal's high standards. |
|
Comments 1: Firstly the mechanistic claims are still… too causal! Your response indicates that mechanistic language was softened, yet the title, abstract, results, discussion, and conclusions still make causal pathway-level claims (eg through activation of PINK1/Parkin-mediated mitophagy and by regulating the PINK1/Parkin pathway). Without functional perturbation (genetic/pharmacologic inhibition, flux assays), such wording remains overstated… Associative wording must consistently replace causal phrasing throughout. |
|
Response 1: Thank you very much for your valuable suggestion. You pointed out again that our mechanistic claims regarding the role of the PINK1/Parkin pathway should be tempered in the absence of functional perturbation experiments. We fully agree with this suggestion. We have softened our language throughout the manuscript to reflect the associative nature of the evidence(such as “activation of mitophagy-related markers or mitophagy markers ” instead of “through activation of PINK1/Parkin-mediated mitophagy and by regulating the PINK1/Parkin pathway”). The title has been revised to: "Acteoside Ameliorates Diet-Induced Metabolic Dysfunction-Associated Steatotic Liver Disease in Mice By Improving Mitophagy". Mention exactly where in the revised manuscript this change can be found – page 1, line 2-4, 22, page 2, line 68, and page 13, line 399, 413. |
|
Comments 2: The manuscript currently lacks critical methodological details expected for animal research. The randomization procedure is not described in operational terms: •There is no statement that histology, TEM or WB/IHC assessments were performed under blinded conditions •There is no clarification on whether a sample size calculation was done or that n=8 was chosen pragmatically •The rationale for using only male C57BL/6J mice is not written in the methods •Housing is described but enrichment and predefined humane endpoints are not mentioned These omissions weaken the credibility of the experimental design and must be incorporated…. Really! |
|
Response 2: We sincerely thank the reviewer for the thorough assessment and valuable suggestions regarding the methodological reporting in our manuscript. We have carefully revised the manuscript to address all the points raised, as detailed below: • We agree and apologize for this omission. The following statement has been added to the respective methodological sections(page 4, lines 137, 163, and 173): The histopathological evaluations, including hepatocyte injury, inflammatory infiltration, and steatosis, were performed in a blinded manner by independent specialists using a light microscope. Signals were developed with ECL, exposed to X-ray film, and quantified in a blinded manner by independent specialists using ImageJ (v1.46r) with GAPDH normalization. The quantitative analysis of target protein expression was blindly performed by independent specialists using ImageJ software (Version 1.46r), with GAPDH as the internal reference. •This is a valid point. We have added a justification in the Animal Experiments section(page 3, lines 106-110): We selected the group size of n=8 was based on our preliminary experimental data and common practice in similar animal studies of metabolic liver disease, while also adhering to the principles of the 3Rs (Replacement, Reduction, Refinement) in animal research. This sample size is consistent with previously published studies utilizing comparable experimental designs [17,19]. •Thank you for highlighting this. We have added the rationale as follows(page 4, lines 92-96): We chose the strain and sex in line with our previous study and common practice in metabolic liver disease research. This model exhibits a robust development of diet-induced obesity and fatty liver phenotypes, exhibiting shorter modeling duration, higher modeling consistency, and lower variability compared to females[17,18]. •We appreciate this suggestion. The following details have been added as follows(page 3, lines 117-118): At the end of the treatment period, all mice were euthanized via pentobarbital sodium anesthesia after fasted for 12 hours. Mention exactly where in the revised manuscript this change can be found – page 3, line 88-97. |
|
Comments 3: While the statistical section has improved, critical information remains absent: •It is not specified which outcomes required non-parametric testing. •The experimental unit (mouse) must be stated explicitly. •A central statement clarifying sample size per outcome type is missing: •n=8 for biochemical/histologic outcomes •n=3 for WB/IHC (and acknowledged as a limitation) •The term “dose-dependent” is still used despite several non-monotonic results. This should be replaced or formally tested using a trend analysis. |
|
Response 3: Thank you for your attention to the statistical analysis methods in our study. We have supplemented the description of statistical methods in detail following your suggestions and rigorously refined the relevant expressions. The following details have been added to the Statistical Analysis section(page 4, lines 176-187): All statistical analyses were performed using SPSS version 26.0 (IBM Corp., Armonk., NY, USA). The experimental unit for all analyses was the individual mouse, with group sizes of n=8 for biochemical and histopathological assessments and n=3 for WB and IHC analysis. Although the sample size of n=3 for the latter analyses is relatively small and partly reflects funding limitations, it was determined to be statistically adequate for this experimental context. Data are presented as mean ± standard deviation. Normality was tested utilizing the Shapiro-Wilk test and homogeneity of variances was assessed with Levene's test. For normally distributed data with equal variances, one-way ANOVA followed by Tukey’s post hoc test was used for comparisons of the multiple groups. When data did not meet normality or homogeneity assumptions, the Kruskal-Wallis test with Dunn’s post hoc test was utilized(ALT, IL-1β and MDA). Statistical significance was defined as p < 0.05. We have removed the term "dose-dependent" throughout the manuscript and replaced it with more accurate descriptions. |
|
Comments 4: Another issue is the MASLD framing vs actual phenotyping!! The model is framed as MASLD, yet systemic metabolic characterization is limited (no fasting glucose/insulin, HOMA-IR or expanded metabolic profile)!! I explicitly requested either additional data or a clear acknowledgment that the model primarily reflects hepatic steatosis and inflammatory injury, not full multisystem metabolic dysfunction. This has not yet been stated clearly enough. |
|
Response 4: We sincerely thank the reviewer for the insightful comment regarding the alignment between the MASLD framework and the phenotypes characterized in our study. We have supplemented the manuscript with metabolic data, including fasting blood glucose, fasting insulin, and HOMA-IR assessments. These additions have been incorporated into the Methods section (page 3, line 127-132), the Results section (page 7, line 207-215, Figure 2.), and the Discussion(page 12, line 359-370). |
|
Comments 5: Your response promises future work, but the manuscript itself does not explicitly acknowledge that the current imaging analyses are qualitative only… A single sentence recognizing this as a limitation is needed! |
|
Response 5: We thank you for this helpful suggestion. We will perform systematic scoring on the histopathological images and conduct morphometric analyses on the TEM data in future studies. This quantitative approach will provide more robust and objective evidence to reinforce our conclusions. This addition have been incorporated into the Results section (page 13, line 404-407). The morphological assessment of mitochondria in this study was based on qualitative evaluation of TEM images. Although these observations support the main findings, the application of quantitative methodologies in future studies is warranted to generate more objective datasets. |
|
Response to Comments on the Quality of English Language |
|
Point 1: Although language improved, at least one previously flagged phrase remains (“represents is”), and a few expressions (eg dose-dependent pro-protective effects) still require refinement. One more targeted language pass is advisable… |
|
Response 1: We sincerely thank you for your valuable comments on the language issues, which have greatly improved the clarity and quality of our manuscript. We have thoroughly revised the manuscript again. Corrected the grammatical error "represents is" to "represents" or "is" based on the context (page 2, line 57). Conducted a full English language edit to eliminate grammatical errors and improve overall readability. |
|
Special thanks to you for your good comments. Sincerely, Mei-li Cong, PhD. Tao Liu, PhD. Professor School of Public Health, Xinjiang Medical University, Urumqi, 830011, China. Jun Zhao, PhD Xinjiang Key Laboratory for Uighur Medicine, Institute of Materia Medica of Xinjiang, Urumqi 830004, China |